# Maternal prescribed opioid analgesic use during pregnancy and associations with adverse birth outcomes: A population-based study

Ayesha C. Sujan[1]*, Patrick D. Quinn[2], Martin E. Rickert[1], Kelsey K. Wiggs[1], Paul Lichtenstein[3], Henrik Larsson[3,4], Catarina Almqvist[3,5], A. Sara Öberg[3,6], Brian M. D'Onofrio[1,3]

1 Department of Psychological & Brain Sciences, Indiana University Bloomington, Bloomington, Indiana, United States of America, 2 Department of Applied Health Science, School of Public Health, Indiana University Bloomington, Bloomington, Indiana, United States of America, 3 Department of Medical Epidemiology and Biostatistics, Karolinska Institutet, Stockholm, Sweden, 4 School of Medical Sciences, Örebro University, Örebro, Sweden, 5 Astrid Lindgren Children's Hospital, Karolinska University Hospital, Stockholm, Sweden, 6 Department of Epidemiology, T.H. Chan School of Public Health, Harvard, Boston, United States of America

* asujan@indiana.edu

**Data Availability Statement:** Because of Swedish laws, the research team cannot share the database used for the current paper, which was created by

## Abstract

### Background

Published research on prescribed opioid analgesic (POA) use during pregnancy and birth outcomes is limited in scope and has not adequately adjusted for potential confounding factors. To help address these gaps, we estimated associations between maternal POAs during pregnancy and two adverse birth outcomes using a large population-based dataset, multiple definitions of POA exposure, and several methods to evaluate the influence of both measured and unmeasured confounding factors.

### Methods and findings

We obtained data by linking information from several Swedish registers and conducted a retrospective cohort study on a population-based sample of 620,458 Swedish births occurring between 2007 and 2013 (48.6% female; 44.4% firstborn). We evaluated associations between prenatal POA exposure and risk for preterm birth (PTB; <37 gestational weeks) and small for gestational age (SGA; birth weight 2 standard deviations below the expected weight for gestational age or lower). We evaluated the influence of confounding by adjusting for a wide range of measured covariates while comparing exposed and unexposed infants. Additionally, we adjusted for unmeasured confounding factors by using several advanced epidemiological designs. Infants exposed to POAs anytime during pregnancy were at increased risk for PTB compared with unexposed infants (6.4% exposed versus 4.4% unexposed; adjusted odds ratio [OR] = 1.38, 95% confidence interval [CI] 1.31–1.45, $p < 0.001$). This association was attenuated when we compared POA-exposed infants with acetaminophen-exposed infants (OR = 1.18, 95% CI 1.07–1.30, $p < 0.001$), infants born to women

linking several Swedish registers. However, other researchers may request to gain access to the registers through Statistics Sweden, a Swedish government agency (https://www.scb.se/en/).

**Funding:** Research reported in this publication was supported by a National Science Foundation Graduate Research Fellowship (1342962 [Principle investigator: ACS]; https://www.nsf.gov), the National Institute on Drug Abuse of the National Institutes of Health (R01DA048042 [Principle investigators: BMD and ASO] and R00DA040727 [Principle investigator: PDQ]; https://www.drugabuse.gov), the Swedish Initiative for Research on Microdata in the Social and Medical Sciences (SIMSAM) framework (340-2013-5867 [Principle investigator: CA]; https://simsam.nu), the Swedish Research Council Health, Working Life and Welfare (FORTE; 50623213 [Principle investigator: PL]; https://www.government.se/government-agencies/swedish-research-council-for-health-working-life–forskningsradet-for-arbetsliv-halsa-och-valfard-forte/), and the Swedish Research Council (2014-38313831 [Principle investigators: CA and BMD] and 2018-02679 [Principle investigators: BMD, ASO, and PL]; https://www.vr.se/english.html). The funders did not play any role in the study design, data collection and analysis, decision to publish, or preparation of the manuscript.

**Competing interests:** I have read the journal's policy and the authors of this manuscript have the following competing interests: PDQ declared that research reported in this publication was supported by the National Institute on Drug Abuse of the National Institutes of Health under award number R00DA040727; BMD declared that he has received grants from NIH to support the work; HL reported having served as a speaker for Eli Lilly and Shire and receiving a research grant from Shire, all outside the submitted work; and PL reported having served as a speaker for Medice. The authors have declared that no other competing interests exist.

**Abbreviations:** ATC, Anatomical Therapeutic Chemical; CI, confidence interval; ICD, International Classification of Disease; LMP, last menstrual period; OR, odds ratio; POA, prescribed opioid analgesic; PTB, preterm birth; SGA, small for gestational age.

who used POAs before pregnancy only (OR = 1.05, 95% CI 0.96–1.14, $p$ = 0.27), and unexposed siblings (OR = 0.99, 95% CI 0.85–1.14, $p$ = 0.92). We also evaluated associations with short-term versus persistent POA use during pregnancy and observed a similar pattern of results, although the magnitudes of associations with persistent exposure were larger than associations with any use or short-term use. Although short-term use was not associated with SGA (adjusted $OR_{single-trimester}$ = 0.95, 95% CI 0.87–1.04, $p$ = 0.29), persistent use was associated with increased risk for SGA (adjusted $OR_{multiple-trimester}$ = 1.40, 95% CI 1.17–1.67, $p$ < 0.001) compared with unexposed infants. The association with persistent exposure was attenuated when we used alternative comparison groups (e.g., sibling comparison OR = 1.22, 95% CI 0.60–2.48, $p$ = 0.58). Of note, our study had limitations, including potential bias from exposure misclassification, an inability to adjust for all sources of confounding, and uncertainty regarding generalizability to countries outside of Sweden.

## Conclusions

Our results suggested that observed associations between POA use during pregnancy and risk of PTB and SGA were largely due to unmeasured confounding factors, although we could not rule out small independent associations, particularly for persistent POA use during pregnancy.

## Author summary

### Why was this study done?

- Many pregnant women experience pain during pregnancy, which can be treated with opioids.
- However, the consequences of using opioid medications to treat pain during pregnancy are unclear.
- We therefore evaluated whether opioids used to treat pain during pregnancy increase the risk of two adverse birth outcomes, specifically preterm birth and fetal growth restriction.

### What did the researchers do and find?

- We used several methods to account for background differences between infants born to women treated with opioid analgesics during pregnancy and unexposed infants.
- Our results showed that the absolute risks of the outcomes were low following opioid analgesic treatment during pregnancy and suggested that observed associations between opioid analgesic treatment during pregnancy and the adverse birth outcomes were largely due to background factors rather than exposure to the medication itself.

**What do these findings mean?**

- The findings have important clinical implications as they could help doctors and patients better weigh the risks and benefits of opioid treatment during pregnancy.

- The results also indicate that women of childbearing age should be screened for a broad range of risk factors, and interventions aimed at reducing the incidence of adverse birth outcomes associated with maternal opioid treatment during pregnancy should target co-occurring risk factors.

## Introduction

While estimates of the prevalence of opioid prescriptions among pregnant women vary across studies, research suggests that a substantial proportion of women are treated with opioids during pregnancy [1–5]. For example, a study of pregnant women enrolled in Medicare in the United States reported that 30% of pregnant women fill prescriptions for opioids [5]. Given that opioids cross the human placental barrier, maternal use of these medications results in fetal exposure [6]. However, effects of prenatal exposure to both illicit and prescription opioids on child development are unclear [7–9]. The existing research on opioid use during pregnancy has largely focused on illicit opioids (e.g., heroin) or buprenorphine and methadone in the context of medication-assisted treatment for opioid use disorder [10–14]. However, the use of prescribed opioid analgesics (POAs) for treatment of pain is much more common than illicit use of opioids [15] or medication-assisted treatment [16]. For example, approximately 5% of women use an illicit substance during pregnancy in the US [15], and less than 3% of women prescribed opioids are prescribed these medications to treat opioid use disorder [16].

The limited research on POA use during pregnancy has focused on birth outcomes, particularly preterm birth (PTB), fetal growth, and structural birth defects [7–9], all of which have potential consequences for future morbidity and mortality [17–19]. Some observational studies have reported statistically significant associations between prenatal POA exposure and birth outcomes, whereas others have not [7–9]. Moreover, it is unclear if observed associations are due to causal effects of POA exposure or confounding from indications for maternal analgesic use (e.g., traumatic injury, acute or chronic inflammation, musculoskeletal or neuropathic pain) or other patient characteristics (e.g., psychiatric disorders, concurrent use of other psychiatric medication) [16,20–25]. To help address these gaps, we estimated associations between maternal POAs during pregnancy and PTB and small for gestational age (SGA; a proxy for fetal growth restriction) in a large, population-based dataset using multiple definitions of POA exposure and several methods to evaluate the influence of both measured and unmeasured confounding factors. To our knowledge, no previous study has used methods to evaluate the influence of unmeasured confounding, which is critical because solely adjusting for measured characteristics is unlikely to adequately capture all sources of confounding [26,27]. Because no single observational method can account for all plausible confounding factors, we sought converging evidence from multiple comparisons to rigorously test causal hypotheses [28–30]. Specifically, we compared POA-exposed infants with infants exposed to acetaminophen (i.e., paracetamol), infants born to mothers with POA prescriptions before but not during pregnancy, and their unexposed siblings.

## Methods

We conducted a retrospective cohort, complete-case analysis study on a population-based sample of 620,458 Swedish births from July 1, 2007, to December 31, 2013, who were not exposed to opioids for the treatment of opioid use disorder. We processed and analyzed data using SAS 9.4 and STATA 15.1. A STROBE checklist (S1 Appendix) and a description of our planned analyses (S2 Appendix) are included in the supplemental materials.

The institutional review board at Indiana University and the regional ethical review board in Stockholm, Sweden, approved this study. The study used data available from national registers, and informed consent was not necessary.

We obtained data by linking information from several Swedish registers [31]. The Multi-Generation Register includes biological relationships for all individuals born from 1932 and residing in Sweden since 1961 [32]. The Swedish Prescribed Drug Register includes records of filled medication prescriptions since July 2005 [33,34]. The Medical Birth Register includes information on 96%–99% of births and pregnancy characteristics since 1973 [35–37]. The National Patient Register includes diagnostic codes from all hospital admissions since 1987 and 80% of specialist outpatient care since 2001 [38,39]. The National Crime Register includes criminal convictions since 1973 [40,41]. The Education Register includes highest level of completed formal education [42]. The integrated database for labor market research includes annual socioeconomic data for all individuals since 1990 [43].

### Measures

**POA exposures.** S3 Appendix provides detailed information on the included POAs, including frequencies of prescriptions of specific POA medications and information on the type of clinic prescribing the medications. For example, while many prescriptions originated from obstetrics/gynecology and maternity care clinics, the majority of prescriptions originated from other types of clinics.

We included all prescriptions with Anatomical Therapeutic Chemical (ATC) codes beginning with N02A, acetaminophen/codeine combination medications, with ATC code N02BE51, and buprenorphine (N02AE01, N07BC01, N07BC51) and methadone (N07BC02) prescribed as POAs. We distinguished buprenorphine and methadone for pain treatment versus opioid use disorder treatment with criteria used in previous publications [44,45].

We considered several exposure definitions using maternal POA prescriptions during non-overlapping windows of time (Fig A in S3 Appendix). We defined these windows relative to approximated last menstrual period (LMP) and conception dates. Consistent with prior research [46], we first estimated LMP by subtracting gestational age (predominantly based on ultrasound measurements occurring the 18th to 20th week of pregnancy) from birth date and then estimated conception date by adding 14 days to LMP. Gestational age was predominantly based on ultrasound measurements given that approximately 95% of all pregnant women in Sweden undergo at least one ultrasound [47].

Before-pregnancy-only exposure included those with a prescription 360 days before conception to 91 days before conception and no prescriptions 90 days before conception to birth.

Exposure during a washout period only (i.e., the period in which there would be ambiguity regarding whether a filled prescription would lead to use during pregnancy) included those with a prescription 90 days before conception to the day before conception and no prescriptions from conception to birth. We chose a 90-day window for the washout period because in Sweden the maximum amount of opioid medication dispensed is for a three-month period [48]. Including the washout-period-only variable as a predictor in models prevented

individuals who only had filled prescriptions that occurred shortly before pregnancy from being classified as either exposed or unexposed during pregnancy.

We defined during-pregnancy exposure as filled prescriptions anytime from conception to birth. We also used the timing of maternal prescriptions across trimesters to create proxies for short-term use (occurring in a single trimester) versus persistent use (occurring in multiple trimesters). Any use was the primary exposure, and secondary analyses examined whether associations differed by duration of use. We did not consider timing of exposure as defined by use in specific trimesters because preliminary analyses did not provide support for sensitive periods of exposure during pregnancy (S4 Appendix). Specifically, these preliminary analyses showed that while the magnitude of the associations with first-trimester exposure only (conception to 89 days after LMP) were slightly larger than the magnitude of associations with second-trimester (90 to 179 days after LMP) or third-trimester (180 days after LMP to birth) exposure, the associations were not statistically significantly different for both PTB ($p = 0.25$) and SGA ($p = 0.54$).

**Active comparator medication exposure.** We considered exposures to maternal acetaminophen (N02BE01) as prescriptions (1) anytime during pregnancy, (2) in a single trimester, and (3) in multiple trimesters. For all three exposures, we excluded individuals with maternal POA prescriptions anytime during pregnancy in order to capture acetaminophen-only exposure.

**Covariates.** Pregnancy-related characteristics included birth order, year of birth, maternal-reported smoking during the first trimester, and maternal prescriptions of other psychiatric medications during pregnancy (Table A in S3 Appendix includes ATC codes).

We included maternal and paternal characteristics that were associated with subsequent receipt of POAs [20,49]. Specifically, maternal and paternal characteristics included pre-conception inpatient and outpatient diagnoses of opioid use disorder, non-opioid substance use disorder, schizophrenia or bipolar disorder, and definite or uncertain suicide attempts made according to International Classification of Disease (ICD) criteria; any pre-conception conviction of violent, nonviolent, drug or alcohol, or driving-related crimes; age at childbirth; highest level of education at year of birth; and country of origin.

Other familial and socioeconomic characteristics included maternal reports of parents cohabitating at birth, family-level income quintile at year of birth relative to the Swedish population in that year, and neighborhood deprivation score quintile at year of birth relative to the Swedish population in that year. Neighborhood deprivation score was based on a principal component analysis of several yearly indicators for small geographical areas constructed to delineate socially homogenous areas. The deprivation score for a given area incorporated the proportion of welfare recipients, unemployed individuals, immigrants, divorced individuals, and individuals with low educational attainment, as well as measures of residential mobility, crime rates, and neighborhood disposable income [50].

**Outcomes.** Outcomes were PTB (birth before 37 gestational weeks) and SGA (birth weight 2 standard deviations below the expected fetal weight for gestational age or lower).

## Associations between POA exposure and birth outcomes

We fit five logistic regression models estimating associations between POA exposure and birth outcomes in the analytic sample using robust standard errors to account for clustering of individuals within nuclear families bound by the same biological mothers (i.e., siblings). In all models we included washout-period-only POA exposure as a dummy code.

We first fit all models using POA exposure anytime during pregnancy as the predictor. Then, we re-estimated the models using single-trimester and multiple-trimester exposure as predictors. S5 Appendix lists predictors and comparison groups for the main analytic models.

**Population-wide associations comparing exposed with all unexposed infants.** Models 1 and 2 estimated population-wide associations with POA exposure compared with unexposed infants. Model 1 (unadjusted) did not include any covariates. Model 2 (adjusted) included all measured characteristics as covariates in the regression models. We did not use propensity scores to adjust for measured characteristics because with a relatively limited number of covariates, propensity score methods make exactly as much adjustment as more traditional, outcome regression methods [51], and prominent researchers have expressed serious concerns about the causal inferences researchers have drawn based on propensity score methods (e.g., [51,52]). For the purpose of transparency, all parameter estimates from model 2 are listed in S6 Appendix.

**Associations using alternative comparison groups.** Models 3 through 5 used alternative comparison groups consisting of infants likely to share some maternal characteristics with exposed infants. Model 3 (comparative safety) [53] compared POA-exposed infants with acetaminophen-only-exposed infants to assess the relative safety of the two medications. Given that both POAs and acetaminophen are prescribed for the treatment of pain, a null difference between POA-exposed and acetaminophen-exposed infants would indicate that POA use is as safe as a commonly used pain medication and provide support for confounding by indication.

Model 4 (before-pregnancy use) [54] compared infants exposed to POAs during pregnancy with infants born to women with POA prescriptions before but not during pregnancy. By design, model 4 accounted for all unmeasured factors shared by women who were prescribed POAs around the time of pregnancy. No association between during-pregnancy exposure compared with exposure before pregnancy only would suggest that shared factors rather than a causal influence of intrauterine POA exposure is responsible for an observed increased occurrence in adverse birth outcomes among infants born to women who filled POA prescriptions during pregnancy.

Model 5 (sibling comparison) [55] compared POA-exposed infants with their siblings who were unexposed to POAs during pregnancy. By design, this within-family comparisons accounts for all unmeasured genetic and environmental factors that make siblings similar, including all familial factors that were stable across pregnancies. A lack of association in a sibling comparison would suggest that familial risk factors rather than a causal effect of POA exposure during pregnancy explain observed population-wide associations. In addition to adjusting for potential unmeasured confounding factors by design (i.e., factors shared by the comparison groups), models 3 through 5 also included all measured covariates that varied between comparison groups.

### Sensitivity analyses

We also conducted several sensitivity analyses to evaluate our modeling assumptions and the extent to which the exposure and outcome definitions influenced the main analyses results (S7 to S13 Appendix).

**Bias from exposure definitions.** First, to explore potential bias from exposure misclassification, we estimated population associations while adjusting for all the covariates using several different exposure definitions: (1) an expanded exposure window that included the 90 days before conception in case a prescription that was filled shortly before pregnancy was used during pregnancy, (2) a restricted exposure window that excluded the three days before birth in case a woman was prescribed POAs at the end of her pregnancy to use after delivery, and (3) an exposure defined according to filled prescriptions or maternal-reported use in order to capture women who filled POA prescription before pregnancy but used them during pregnancy (S7 Appendix).

Second, to evaluate whether a specific type of POA medication was largely driving our findings, we re-estimated adjusted associations in (1) a subsample excluding births occurring to women with during-pregnancy prescriptions of dextropropoxyphene (N02AC04 and N02AC54) because this medication is no longer prescribed in Sweden and (2) a subsample excluding births occurring to women with during-pregnancy methadone or buprenorphine prescriptions in case these infants were exposed to prescribed opioids for the treatment of opioid use disorder rather than pain (S8 Appendix).

Third, in order to evaluate whether exposure to medications other than POAs that are included in POA combination medications influenced the main analyses result, we estimated adjusted associations in a subsample excluding births occurring to mothers with filled prescriptions of combination POA medications (i.e., oxycodone/naloxone, buprenorphine/naloxone, morphine/antispasmodics, ketobemidone/antispasmodics, hydromorphone/antispasmodics, codeine combinations, and dextropropoxyphene combinations; S9 Appendix).

Fourth, to further assess whether exposure to polypharmacy was responsible for observed associations with POA exposure, we re-estimated adjusted associations in a subsample excluding births occurring to women prescribed other psychiatric medications during pregnancy (S10 Appendix).

**Sibling comparison assumptions.** We conducted two sets of analyses to evaluate the assumptions of sibling comparisons (S11 Appendix). To evaluate whether individuals with siblings differed from the full population, we re-estimated adjusted associations in a subsample comprised of the women that contributed more than once to the full population sample (i.e., women that had more than one singleton birth in the study period). To evaluate if the sibling comparisons were biased by carryover effects (i.e., exposure in an earlier pregnancy affecting subsequent pregnancies), we compared differentially exposed pairs of firstborn cousins.

**Bias from outcome definitions.** To evaluate whether any potential influence of POA exposure was not reflected in the clinical cutoff values of the birth outcomes, we fit all of the main models predicting birth outcomes on continuous scales. Specifically, we predicated (1) gestational age measured in days and (2) birth weight measured in grams adjusted for gestational age (S12 Appendix).

**Influence of missing data.** We conducted two sensitivity analyses to evaluate whether missing data influenced our findings (S13 Appendix). First, we estimated the association between absence of data and POA exposure in the target sample.

Then, we assessed the potential confounding influence of the covariates with missing data by removing these covariates from the fully adjusted model in the analytic sample and fitting this alternative model in both the analytic and target samples to evaluate whether the findings in the restricted (analytic) sample appear to represent the findings of the full (target) sample.

## Results

### Participants

We started with a population-based sample of 711,986 births occurring in Sweden between July 1, 2007, and December 31, 2013. To create the target sample, we sequentially excluded infants with invalid child identifiers (2,648), maternal identifiers (107), and sex (6); multiple births (19,844); births missing gestational age (173); and births exposed to buprenorphine or methadone for opioid use disorder treatment rather than pain (276). To create the analytic sample from the target sample, we excluded 68,474 infants with missing covariate data. The final analytic cohort of 620,458 births represented approximately 90% of the target sample. We

also identified 288,995 births occurring to the same mother in the analytic sample for inclusion in sibling comparison analyses. This sample of siblings that shared mothers included 9,201 unique mothers.

## Demographics

Table 1 compares background characteristics among infants exposed to any POA during pregnancy and all the unexposed infants in the target sample. Additionally, S14 Appendix shows the prevalence of background characteristics stratified by all exposure groups, including infants with single- and multiple-trimester exposure. We provided demographic data from the target sample in order to document the prevalence of missing data on covariates, which ranged from none to approximately four percent.

In the final analytic sample, 4.4% of infants were exposed to POAs anytime during pregnancy, with 3.7% exposed in a single trimester and 0.7% exposed in multiple trimesters (Table 2).

PTB occurred more often among exposed (6.4%) than unexposed (4.4%) infants, particularly among infants with multiple-trimester exposure (5.8% of single-trimester-exposed, 9.6% of multiple-trimester-exposed); and about 2% of infants were SGA regardless of exposure status (2.1% of unexposed, 2.2% of exposed, 2.0% of single-trimester-exposed, and 3.04% of multiple-trimester-exposed; S15 Appendix).

## Associations with PTB

**Exposure anytime during pregnancy.** The unadjusted association between any exposure to POAs and PTB (odds ratio [OR] = 1.48, 95% confidence interval [CI] 1.41–1.56, $p < 0.001$; Table 3) was attenuated but remained robust after adjustment for measured covariates (OR = 1.38, 95% CI 1.31–1.45, $p < 0.001$) and was further attenuated when we used the alternative comparison groups (comparative safety OR = 1.18, 95% CI 1.07–1.30, $p < 0.001$; before-pregnancy-use OR = 1.05, 95% CI 0.96–1.14, $p = 0.27$; siblings OR = 0.99, 95% CI 0.85–1.14, $p = 0.92$).

**Short-term versus persistent exposure during pregnancy.** We observed a similar pattern of attenuated associations with increasing confounder control also when separating single- and multiple-trimester exposure, although this also revealed that associations for multiple-trimester exposure were consistently more pronounced than for single-trimester exposure. The unadjusted associations (single-trimester OR = 1.34, 95% CI 1.26–1.41, $p < 0.001$ and multiple-trimester OR = 2.31, 95% CI 2.08–2.55, $p < 0.001$) suggested a relationship with duration of use that remained despite attenuation in adjusted models (single-trimester OR = 1.27, 95% CI 1.20–1.34, $p < 0.001$ and multiple-trimester OR = 1.97, 95% CI 1.77–2.18, $p < 0.001$); however, the associations were partially attenuated when using some of the alternative comparison groups (comparative safety: single-trimester OR = 0.91, 95% CI 0.83–0.99, $p = 0.03$ and multiple-trimester OR = 1.53, 95% CI 1.36–1.72, $p < 0.001$; before-pregnancy-use: single-trimester OR = 1.00, 95% CI 0.92–1.09, $p = 0.99$ and multiple-trimester OR = 1.52, 95% CI 1.28–1.80, $p < 0.001$) and fully attenuated in the sibling comparison (single-trimester OR = 0.99, 95% CI 0.85–1.15, $p = 0.87$ and multiple-trimester OR = 1.04, 95% CI 0.70–1.55, $p = 0.83$).

## Associations with SGA

**Exposure anytime during pregnancy.** Across models 1 through 5, associations were negligible (Table 3).

**Short-term versus persistent exposure during pregnancy.** Similar to the findings for PTB, the association between POAs exposure and SGA differed according to the duration of

**Table 1. Demographics in the target sample.**

| Pregnancy-related characteristics | Unexposed | Ever exposed | Associations comparing exposed to unexposed | |
|---|---|---|---|---|
| | (*n* = 658,580 [95.59%]) | (*n* = 30,352 [4.41%]) | | |
| | N (%) | N (%) | OR (95% CI) | *p*-value |
| Birth order | | | | |
| 1st | 296,455 (45.01) | 11,883 (39.15) | Reference | Reference |
| 2nd | 241,854 (36.72) | 10,884 (35.86) | 1.12 (1.09–1.15) | <0.001 |
| 3rd or higher | 120,271 (18.26) | 7,585 (24.99) | 1.57 (1.53–1.62) | <0.001 |
| Year of birth | | | | |
| 2007 to 2009 | 246,792 (37.47) | 11,016 (36.29) | Reference | Reference |
| 2010 to 2013 | 411,788 (62.53) | 19,336 (63.71) | 1.05 (1.03–1.08) | <0.001 |
| Maternal smoking during the first trimester | | | | |
| None | 596,358 (90.55) | 25,881 (85.27) | Reference | Reference |
| Moderate (1 to 9 cigarettes per day) | 30,513 (4.63) | 2,455 (8.09) | 1.85 (1.78–1.94) | <0.001 |
| High (10 or more cigarettes per day) | 8,409 (1.28) | 918 (3.02) | 2.52 (2.35–2.70) | <0.001 |
| Missing | 23,300 (3.54) | 1,098 (3.62) | 1.09 (1.02–1.16) | 0.01 |
| Exposure to other psychiatric medications | 28,432 (4.32) | 4,695 (15.47) | 3.74 (3.63–3.86) | <0.001 |
| **Maternal characteristics** | | | | |
| Opioid use disorder before conception | 468 (0.07) | 123 (0.41) | 5.72 (4.69–6.98) | <0.001 |
| Non-opioid substance use disorder before conception | 10,298 (1.56) | 960 (3.16) | 2.06 (1.92–2.20) | <0.001 |
| Schizophrenia or bipolar disorder before conception | 2,576 (0.39) | 285 (0.94) | 2.42 (2.14–2.73) | <0.001 |
| Definite or uncertain suicide attempt before conception | 11,181 (1.70) | 1,247 (4.11) | 2.48 (2.34–2.63) | <0.001 |
| Any criminal convictions before conception | 43,190 (6.56) | 3,352 (11.04) | 1.77 (1.70–1.84) | <0.001 |
| Age at year of birth | | | | |
| Less than 19 years | 6,797 (1.03) | 188 (0.62) | 0.62 (0.53–0.71) | <0.001 |
| 20 to 29 years | 256,894 (39.01) | 11,526 (37.97) | Reference | Reference |
| 30 to 39 years | 362,436 (55.03) | 16,858 (55.54) | 1.04 (1.01–1.06) | <0.001 |
| 40 to 45 years | 31,600 (4.80) | 1,736 (5.72) | 1.22 (1.16–1.29) | <0.001 |
| 46 years and older | 853 (0.13) | 44 (0.14) | 1.15 (0.85–1.56) | 0.37 |
| Highest level of education at year of birth | | | | |
| Less than 9 years | 19,895 (3.02) | 865 (2.85) | Reference | Reference |
| 9 years | 50,739 (7.70) | 3,729 (12.29) | 1.69 (1.57–1.82) | <0.001 |
| 1 to 3 years upper secondary | 239,629 (36.39) | 13,022 (42.90) | 1.25 (1.17–1.34) | <0.001 |
| Any postsecondary or postgraduate | 332,690 (50.52) | 12,301 (40.53) | 0.85 (0.79–0.91) | <0.001 |
| Missing | 15,629 (2.37) | 435 (1.43) | 0.64 (0.57–0.72) | <0.001 |
| Country of origin | | | | |
| Sweden | 502,636 (76.32) | 24,079 (79.33) | 1.19 (1.16–1.23) | <0.001 |
| Missing | 98 (0.01) | 1 (0.00) | 0.26 (0.04–1.82) | 0.17 |
| **Paternal characteristics** | | | | |
| Opioid use disorder before conception | 984 (0.15) | 102 (0.34) | 2.26 (1.84–2.77) | <0.001 |
| Non-opioid substance use disorder before conception | 11,554 (1.75) | 832 (2.74) | 1.58 (1.47–1.70) | <0.001 |
| Schizophrenia or bipolar disorder before conception | 1,968 (0.30) | 144 (0.47) | 1.59 (1.34–1.88) | <0.001 |
| Definite or uncertain suicide attempt before conception | 7,980 (1.21) | 474 (1.56) | 1.29 (1.18–1.42) | <0.001 |
| Any criminal convictions before conception | 120,387 (18.28) | 7,117 (23.45) | 1.37 (1.33–1.41) | <0.001 |
| Age at year of birth | | | | |
| Less than 19 years | 2,023 (0.31) | 67 (0.22) | 0.71 (0.56–0.91) | 0.01 |
| 20 to 29 years | 157,717 (23.95) | 7,352 (24.22) | Reference | Reference |
| 30 to 39 years | 377,775 (57.36) | 17,021 (56.08) | 0.97 (0.94–0.99) | 0.02 |

(*Continued*)

**Table 1.** (Continued)

| Pregnancy-related characteristics | Unexposed | Ever exposed | Associations comparing exposed to unexposed | |
|---|---|---|---|---|
| | (*n* = 658,580 [95.59%]) | (*n* = 30,352 [4.41%]) | | |
| | N (%) | N (%) | OR (95% CI) | *p*-value |
| 40 to 45 years | 79,403 (12.06) | 3,939 (12.98) | 1.06 (1.02–1.11) | <0.001 |
| 46 years and older | 27,724 (4.21) | 1,346 (4.43) | 1.04 (0.98–1.11) | 0.18 |
| Missing | 13,938 (2.12) | 627 (2.07) | 0.97 (0.89–1.05) | 0.40 |
| Highest level of education at year of birth | | | | |
| Less than 9 years | 17,865 (2.71) | 788 (2.60) | Reference | Reference |
| 9 years | 60,313 (9.16) | 3,432 (11.31) | 1.29 (1.19–1.40) | <0.001 |
| 1 to 3 years upper secondary | 293,777 (44.61) | 15,014 (49.47) | 1.16 (1.08–1.25) | <0.001 |
| Any postsecondary or postgraduate | 259,281 (39.37) | 9,985 (32.90) | 0.87 (0.81–0.94) | <0.001 |
| Missing | 27,344 (4.15) | 1,133 (3.73) | 0.94 (0.86–1.03) | 0.19 |
| Country of origin | | | | |
| Sweden | 492,997 (74.86) | 23,367 (76.99) | 1.13 (1.10–1.16) | <0.001 |
| Missing | 14,042 (2.13) | 629 (2.07) | 1.07 (0.98–1.16) | 0.12 |
| **Other familial and socioeconomic characteristics** | | | | |
| Parental cohabitation status at birth | | | | |
| Parents not cohabiting at birth | 39,845 (6.05) | 2,302 (7.58) | 1.28 (1.22–1.34) | <0.001 |
| Missing | 24,936 (3.79) | 1,194 (3.93) | 1.06 (1.00–1.12) | 0.06 |
| Family income at year of birth | | | | |
| 1st quintile (lowest income) | 55,036 (8.36) | 2,183 (7.19) | 0.83 (0.79–0.87) | <0.001 |
| 2nd quintile | 85,747 (13.02) | 4,319 (14.23) | 1.03 (0.99–1.07) | 0.11 |
| 3rd quintile | 190,101 (28.87) | 9,292 (30.61) | Reference | Reference |
| 4th quintile | 222,018 (33.71) | 10,098 (33.27) | 0.93 (0.90–0.96) | <0.001 |
| 5th quintile (highest income) | 105,678 (16.05) | 4,460 (14.69) | 0.86 (0.83–0.90) | <0.001 |
| Missing | 2,005 (0.30) | 39 (0.13) | 0.48 (0.35–0.66) | <0.001 |
| Neighborhood deprivation at year of birth | | | | |
| 1st quintile (least neighborhood deprivation) | 96,203 (14.61) | 4,179 (13.77) | 0.96 (0.92–1.00) | 0.05 |
| 2nd quintile | 113,124 (17.18) | 5,037 (16.60) | 0.98 (0.94–1.02) | 0.35 |
| 3rd quintile (reference) | 118,578 (18.01) | 5,379 (17.72) | Reference | Reference |
| 4th quintile | 140,301 (21.30) | 6,532 (21.52) | 1.03 (0.99–1.07) | 0.24 |
| 5th quintile (most neighborhood deprivation) | 190,374 (28.91) | 9,225 (30.39) | 1.07 (1.03–1.11) | <0.001 |
| Missing | 1,189 (0.18) | 43 (0.14) | 0.83 (0.61–1.13) | 0.24 |

Abbreviations: CI, confidence interval; OR, odds ratio

use (Table 3). While there was no association with single-trimester exposure (unadjusted OR = 0.96, 95% CI 0.88–1.06, *p* = 0.43), a moderate association with multiple-trimester exposure (unadjusted OR = 1.45, 95% CI 1.22–1.73, *p* < 0.001) remained in the adjusted (OR = 1.40, 95% CI 1.17–1.67, *p* < 0.001) and comparative safety (OR = 1.41, 95% CI 1.15–1.73, *p* = 0.001) models but was attenuated in the before-pregnancy-use (OR = 1.20, 95% CI 0.89–1.60, *p* = 0.23) and siblings (OR = 1.22, 95% CI 0.60–2.48, *p* = 0.58) models.

## Sensitivity analyses

**Bias from exposure definitions.** S7 Appendix examined potential bias from exposure misclassification and found the same pattern of results with a number of alternative exposure definitions (i.e., the expanded prescription window definition, the restricted prescription

**Table 2. Proportion of infants with POA and acetaminophen exposure in the analytic sample.**

| Exposure | N (%) |
|---|---|
| **POA exposure** | |
| Before-pregnancy-only | 18,883 (3.04) |
| Washout-period-only | 7,199 (1.16) |
| Anytime during pregnancy | 27,559 (4.44) |
| Single-trimester | 23,211 (3.74) |
| Multiple-trimesters | 4,348 (0.70) |
| **Acetaminophen-only exposure** | |
| Anytime during pregnancy | 13,116 (2.11) |
| Single-trimester | 10,293 (1.66) |
| Multiple-trimesters | 2,823 (0.45) |

Abbreviation: POA, prescribed opioid analgesic

window definition, and the definition according to maternal-reported POA use or during-pregnancy prescriptions). S8 Appendix suggested that findings were not largely driven by dextropropoxyphene, an opioid that is no longer prescribed, or by opioids that are also prescribed for the treatment of opioid use disorder. S9 Appendix showed commensurate adjusted associations in a subsample excluding those with during-pregnancy prescriptions of combination POA medications, suggesting that the results were not driven by inclusion of combination POA medications. Similarly, S10 Appendix showed comparable adjusted associations in a subsample excluding those with during-pregnancy polypharmacy, suggesting that associations with POA exposure observed in main analyses were not driven by exposure to polypharmacy.

**Sibling comparison assumptions.** Two analyses in S11 Appendix examined assumptions of sibling comparisons. The first analysis suggested that attenuation in sibling comparisons was not due to restricting the sample to infants with siblings, because population-wide associations were comparable in samples of infants with and without siblings. The second analysis suggested that sibling comparison findings were not due to carryover effects (i.e., exposure in a prior pregnancy affecting subsequent pregnancies) because comparisons of firstborn cousins showed the same pattern of results as the sibling comparison findings.

**Bias from outcome definitions.** S12 Appendix demonstrated that using clinical cutoff values for the outcomes did not cause a failure to detect an influence of POA exposure by estimating associations with two continuously measured birth outcomes. Any POA exposure was associated with reduced gestational age in unadjusted models, and the associations attenuated in subsequent models. POA exposure was not associated with reduced birth weight adjusted for gestational age across any of the models.

**Influence of missing data.** S13 Appendix suggests that excluding infants with missing data did not bias our results. Absence of data was not associated with prescriptions anytime during pregnancy after adjusting for all measured covariates in our target sample (OR = 1.03, 95% CI 0.98–1.08), indicating that covariate adjustment helped minimize potential bias from missing data. Moreover, removing the covariates with missing data (smoking during pregnancy, maternal and paternal education, maternal and paternal country of origin, paternal age at childbearing, parental cohabitation status at birth, family income, and neighborhood deprivation) from the fully adjusted model did not meaningfully change the association in the analytic sample, and this alternative model yielded similar estimates of association in the target sample as the analytic sample.

**Table 3. Associations between POA exposure and adverse birth outcomes.**

| Exposure and outcomes | Model 1: Unadjusted | | Model 2: Adjusted | | Model 3: Comparative safety | | Model 4: Before-pregnancy-only comparison | | Model 5: Sibling comparison | |
|---|---|---|---|---|---|---|---|---|---|---|
| | OR (95% CI) | *p*-value | OR (95% CI) | *p*-value | OR (95% CI) | *p*-value | OR (95% CI) | *p*-value | OR (95% CI) | *p*-value |
| **PTB** | | | | | | | | | | |
| Exposure anytime during pregnancy | 1.48 (1.41–1.56) | <0.001 | 1.38 (1.31–1.45) | <0.001 | 1.18 (1.07–1.30) | <0.001 | 1.05 (0.96–1.14) | 0.27 | 0.99 (0.85–1.14) | 0.92 |
| Exposure in a single trimester | 1.34 (1.26–1.41) | <0.001 | 1.27 (1.20–1.34) | <0.001 | 0.91 (0.83–0.99) | 0.03 | 1.00 (0.92–1.09) | 0.99 | 0.99 (0.85–1.15) | 0.87 |
| Exposure in multiple trimesters | 2.31 (2.08–2.55) | <0.001 | 1.97 (1.77–2.18) | <0.001 | 1.53 (1.36–1.72) | <0.001 | 1.52 (1.28–1.80) | <0.001 | 1.04 (0.70–1.55) | 0.83 |
| **SGA** | | | | | | | | | | |
| Exposure anytime during pregnancy | 1.04 (0.96–1.13) | 0.35 | 1.02 (0.93–1.10) | 0.64 | 0.98 (0.85–1.13) | 0.80 | 0.93 (0.81–1.06) | 0.26 | 0.91 (0.70–1.19) | 0.55 |
| Exposure in a single trimester | 0.96 (0.88–1.06) | 0.43 | 0.95 (0.87–1.04) | 0.29 | 0.87 (0.76–1.01) | 0.06 | 0.90 (0.78–1.03) | 0.13 | 0.90 (0.69–1.18) | 0.44 |
| Exposure in multiple trimesters | 1.45 (1.22–1.73) | <0.001 | 1.40 (1.17–1.67) | <0.001 | 1.41 (1.15–1.73) | 0.001 | 1.20 (0.89–1.60) | 0.23 | 1.22 (0.60–2.48) | 0.58 |

Models 1 and 2 estimated population-wide associations with POA-exposed infants compared with unexposed infants. Model 1 did not include any covariates. Model 2 included all measured characteristics as covariates in the regression models. Models 3 through 5 used alternative comparison groups consisting of infants likely to share some characteristics with exposed infants while also controlling for measured covariates. Model 3 compared infants born to women prescribed POAs during pregnancy with infants born to women prescribed acetaminophen only during pregnancy. Model 4 compared infants born to women prescribed POAs during pregnancy with infants born to women prescribed POA before but not during pregnancy. Model 5 compared POA-exposed infants with their unexposed siblings.

Abbreviations: CI, confidence interval; OR, odds ratio; POA, prescribed opioid analgesic; PTB, preterm birth; SGA, small for gestational age

## Discussion

In a population-based sample of Swedish births occurring between 2007 and 2013, we used multiple observational designs to evaluate the consequences of prenatal POA exposure on the risk of two adverse birth outcomes—PTB and SGA. Compared with unexposed infants, infants exposed to POAs anytime during pregnancy, in a single trimester, and in multiple trimesters were all at increased risk for PTB, although infants exposed in multiple trimesters had the greatest risk. However, when we used comparison groups consisting of unexposed infants that shared characteristics with exposed infants, the associations with PTB were attenuated, suggesting that observed associations were largely due to unmeasured confounding factors. For example, siblings exposed anytime during pregnancy, in a single trimester, and in multiple trimesters were not at increased PTB risk compared with their unexposed siblings, which suggests that unmeasured genetic and environmental factors shared by siblings account for observed population-wide associations.

For SGA, we only observed associations with persistent POA exposure (i.e., in multiple trimesters). We observed the higher risk of SGA among exposed infants when we made comparisons to unexposed infants, as well as acetaminophen-exposed infants. However, the association was attenuated when we used infants born to mothers with POA prescriptions before pregnancy only and unexposed siblings as the comparison groups, which again suggests that the observed associations were largely due to confounding factors.

Our study had several noteworthy strengths that distinguish it from previous studies. First, we used multiple methods that were able to account for both measured and unmeasured sources of confounding and found converging evidence across these methods that suggested that the observed associations with birth outcomes were at least partially due to confounding.

This converging evidence suggests that our conclusions are not due to violations of the assumptions of one design. Second, we reduced the potential influence from exposure misclassification by including a 90-day prepregnancy washout period and conducting multiple sensitivity analyses to evaluate the influence of potential exposure misclassification. Third, unlike most previous studies that examined any POA use anytime during pregnancy, we defined exposure by the number of trimesters with prescriptions and the number of prescriptions throughout pregnancy, as well as evaluated for sensitive periods of exposure by estimating associations with exposure earlier and later in pregnancy. Fourth, we excluded opioids used to treat opioid use disorder. Most previous studies focused on two opioids—methadone and buprenorphine—in the context of medication-assisted treatment of opioid use disorder or did not differentiate between POAs and opioids prescribed for opioid use disorder treatment. This is an important distinction because women who are prescribed opioids to manage pain presumably differ on several important background characteristics compared with women who are prescribed opioids for the treatment of opioid use disorder.

Our study also had several limitations. First, our exposure was subject to misclassification because mothers may not have taken their prescribed medications. Second, because of the rarity of the exposures, we did not evaluate associations with specific POA medications, and we did not examine pill dosages, number of pills prescribed, and number days covered. Therefore, future research should utilize finer-grained measurements, such as morphine equivalent units, which take into account the number of pills prescribed, numbers of days supplied, strength per pill, and type of POA [56]. Third, the comparative safety model evaluated the relative safety of POA exposure compared with acetaminophen exposure, and a null difference could have been due to adverse effects of both medications rather than confounding by common indications for the medications. Fourth, we did not have measures of indications for POA use, although we used several methods to assess the influence of unmeasured confounding factors. Nonetheless, future research should evaluate associations with POA treatment during pregnancy among women with conditions causing chronic pain (e.g., back pain, abdominal pain, fibromyalgia, and rheumatoid arthritis) [2] in order to compare exposed to unexposed with similar conditions for which POA treatment is indicated. Fifth, we do not know if unmeasured confounding biased results. In particular, we cannot rule out the possibility that further unmeasured confounding factors account for the remaining observed associations, particularly with multiple-trimester exposure. Indeed, women who use POAs in multiple trimesters are likely to have more severe conditions requiring chronic POA treatment, which could result in greater unmeasured confounding by indication. Sixth, we do not know if our findings will generalize to countries outside of Sweden given between-country differences (e.g., higher prescribing rates in the US compared to Sweden) [2,57]; however, in theory these differences would not impact the ability to detect a causal effect of POA exposure. Nonetheless, future research should evaluate whether our findings apply to populations outside of Sweden. Seventh, we excluded infants that did not have information on all covariates. However, sensitivity analyses suggested that excluding these infants from the main analyses did not influence our conclusions (S13 Appendix). Eighth, we only explored two birth outcomes and were not able to explore more rare birth outcomes, such as extreme PTB. However, sensitivity analyses showed the same pattern of results with continuous measures of gestational age and fetal growth (S12 Appendix). Future research should examine the consequences of prenatal POA exposure on other important adverse birth outcomes (e.g., birth defects), as well as outcomes presenting later in development (e.g., autism and attention-deficit hyperactivity disorder).

Despite the limitations, our study has important clinical implications. The vast majority of infants exposed during pregnancy did not have PTB and were not SGA. The absolute risk was low even among infants with persistent exposure, with 90% not having PTB and 97% not

being SGA. Although we could not rule out small independent associations, particularly for persistent exposure during pregnancy, our results suggested that associations between prenatal POA exposure and PTB and SGA were largely due to confounding factors associated with maternal POA use during pregnancy rather than causal effects of POA exposure. We believe our findings are valuable as they may help doctors and patients better weigh the risks and benefits of POA use in women of childbearing age and pregnant women, although decision-making must consider a wide range of potential adverse outcomes. Our results also indicate that women of childbearing years should be assessed for a broad range of risk factors, and interventions aimed at reducing the incidence of adverse birth outcomes associated with maternal POA use during pregnancy should target co-occurring risk factors.

## Supporting information

**S1 Appendix. STROBE checklist.**
(DOCX)

**S2 Appendix. Planned analyses.**
(DOCX)

**S3 Appendix. Additional information on medication exposures.**
(DOCX)

**S4 Appendix. Preliminary analyses exploring for sensitive periods of exposure.**
(DOCX)

**S5 Appendix. Review of models used to evaluate associations between POA use and birth outcomes.** POA, prescribed opioid analgesic.
(DOCX)

**S6 Appendix. All parameter estimates from adjusted models estimating associations with maternal prescribed opioid filled prescriptions anytime during pregnancy.**
(DOCX)

**S7 Appendix. Sensitivity analyses evaluating for potential bias from exposure misclassification.**
(DOCX)

**S8 Appendix. Sensitivity analyses evaluating the influence of type of opioid.**
(DOCX)

**S9 Appendix. Sensitivity analyses evaluating the influence of the inclusion of combination POA medications.** POA, prescribed opioid analgesic.
(DOCX)

**S10 Appendix. Sensitivity analyses evaluating the role of polypharmacy.**
(DOCX)

**S11 Appendix. Sensitivity analyses evaluating assumptions of sibling comparisons results.**
(DOCX)

**S12 Appendix. Sensitivity analyses evaluating associations with continuous outcomes.**
(DOCX)

**S13 Appendix. Sensitivity analyses evaluating the influence of missing data.**
(DOCX)

**S14 Appendix. Prevalence of the covariates stratified by exposure status in the target sample.**
(DOCX)

**S15 Appendix. Prevalence of PTB and SGA among all exposure and comparison groups.**
PTB, preterm birth; SGA, small for gestational age.
(DOCX)

## Acknowledgments

**Disclaimer:** The content is solely the responsibility of the authors and does not necessarily represent the official views of the National Institutes of Health.

## Author Contributions

**Conceptualization:** Ayesha C. Sujan, Patrick D. Quinn, Martin E. Rickert, Kelsey K. Wiggs, Paul Lichtenstein, Henrik Larsson, Catarina Almqvist, A. Sara Öberg, Brian M. D'Onofrio.

**Data curation:** Ayesha C. Sujan, Martin E. Rickert.

**Formal analysis:** Ayesha C. Sujan, Martin E. Rickert.

**Funding acquisition:** Ayesha C. Sujan, Patrick D. Quinn, Paul Lichtenstein, Henrik Larsson, Catarina Almqvist, A. Sara Öberg, Brian M. D'Onofrio.

**Investigation:** Ayesha C. Sujan.

**Methodology:** Ayesha C. Sujan, Patrick D. Quinn, Martin E. Rickert, Kelsey K. Wiggs, Paul Lichtenstein, Henrik Larsson, A. Sara Öberg, Brian M. D'Onofrio.

**Project administration:** Ayesha C. Sujan.

**Resources:** Paul Lichtenstein, Henrik Larsson, Catarina Almqvist, Brian M. D'Onofrio.

**Supervision:** Patrick D. Quinn, A. Sara Öberg, Brian M. D'Onofrio.

**Writing – original draft:** Ayesha C. Sujan.

**Writing – review & editing:** Ayesha C. Sujan, Patrick D. Quinn, Martin E. Rickert, Kelsey K. Wiggs, Paul Lichtenstein, Henrik Larsson, Catarina Almqvist, A. Sara Öberg, Brian M. D'Onofrio.

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
