## [Decision Letter · Decision Letter 0]

7 Sep 2019

Dear Dr. Sujan,

Thank you very much for submitting your manuscript "Maternal prescribed opioid analgesic use during pregnancy and risk for adverse birth outcomes: A population-based study" (PMEDICINE-D-19-02701) for consideration at PLOS Medicine. 

[LINK]

In light of these reviews, I am afraid that we will not be able to accept the manuscript for publication in the journal in its current form, but we would like to consider a revised version that addresses the reviewers' and editors' comments. Obviously we cannot make any decision about publication until we have seen the revised manuscript and your response, and we plan to seek re-review by one or more of the reviewers. 

We expect to receive your revised manuscript by Sep 23 2019 11:59PM. Please email us (plosmedicine@plos.org) if you have any questions or concerns.

We look forward to receiving your revised manuscript. 

Sincerely,

Clare Stone - Acting Editor in Chief

for

Louise Gaynor, MBBS PhD

Associate Editor 

PLOS Medicine

plosmedicine.org

Title – association with, instead of risk, please.

Abstract – paragraph one you say “if observed associations” but don’t tell us what the associations are’; similarly on first mention of the 2 birth outcomes please do say what they are; The abstract is very long, I would suggest cutting the paragraph starting with ‘across all models..’; please provide a sentence on the limitations of the study as the final sentence of the Methods and Findings section of the abstract; please provide some summary demographic information on participants. 

Data – please provide a link or email address for statistics Sweden. 

Throughout inc tables, please provide p values where any data is quantified (already with 95%Cis)

Line 63 – heading please and all other headings should be to the left not central, per house style. Please refer to formatting guidelines.

Refs in the main text need to be in square brackets.

Line 129 – define washout period

Line 252 – dropped. Do you mean excluded? 

Table 2- it is more conventional for the demographic information to be Table 1. Please reorder and amend any call outs in the text to any re-ordered table. 

Line 291 onward, please remove subscript and use normal text (brackets if needed)

Line 385 - missingness – perhaps recast – absence?

Unless I missed it, please do provide the prespecified analysis plan and a STROBE reporting checklist. Both as supp files. 

Comments from the reviewers:

Reviewer #1: This is a well-conducted study on the association between maternal prescribed opioid analgesic use and adverse birth outcomes using population data. The study design, datasets, statistical methods and analyses are mostly adequate and of a good standard, however the presentation and the interpretation of the results could be improved.

1) The abstract is too long especially in the methods and findings part. Need to be concise to summarise the main findings not all findings and also be able to deliver the key messages. At the moment, it is all over the place with too much details therefore a bit difficult to read.

2) Table 2 is a massive table, however it was summarised only by a small paragraph with 5 lines (line 273-277)? and about missing values and their impact? Basically Table 2 is very descriptive, a bit messy to read and not very informative. It would be better to at least have odd ratios in the table to be more informative.

3) Missing data. The study is based on complete data analyses therefore sensitivity analysis of missing data is very important. However, Table 2 is not fit for this purpose. A comparison table between complete data and missing data on all the different characteristics and birth outcomes should be included in the main paper or supplementary information. The differences should be seriously discussed and commented as a way of sensitivity analysis.

4) Table 3 is a key and useful table. However, it needs extensive footnotes to explain clearly what exactly these 5 models were adjusted for. It is not practical and reader-friendly for readers to go back to check these in the methods section.

5) There are quite a few subgroup and sensitivity analyses apart from the main analyses and sometimes it's a bit difficult to follow. Can authors provide clear explanations and logics to why these many sub-analyses are necessary and what they are really for? The same applies to the 5 models.

Reviewer #2: This manuscript is a population-based analysis that aims to determine the causal impact of maternal prescribed opioid analgesic use during pregnancy and adverse birth outcomes. I thought this was an excellent analysis and well-written manuscript, and I was particularly impressed with the series of sensitivity analyses addressing various possible biases and looking for ways to fully account for measured and unmeasured confounding.

Comments:

1. Abstract conclusion, line 60: Would it be fair to say that the confounding factors are those that are usually unmeasured in other studies, e.g. those that were only accounted for when comparing with unexposed siblings or cousins? I think this could be made clearer in the conclusion.

2. Discussion, first paragraph - I think the main findings could be more clearly expressed here. It is a very complex series of analyses, so I understand that they are difficult to condense, but I don't think the main findings, as expressed in the Abstract are clear enough here.

3. Could the authors please further justify their reasons for not dealing with the large proportion of missing data through a principled missing data technique such as multiple imputation or inverse probability weighting? The authors state that missingness was weakly associated with exposure status, but what about the association with the outcome? Were there auxiliary variables available (those not already in the adjusted models) that could have been used in multiple imputation? 

Minor comments:

- I was sometimes confused by the term 'offspring' and would suggest changing this to 'infants' throughout. When reading about 'exposed and unexposed offspring' I thought initially that this was a sibling comparison, as I was interpreting this as the exposed and unexposed infants being the offspring of the same mother.

Line 89 - Acetaminophen is called paracetamol in Australia (and possibly other countries?). I would suggest putting paracetamol in brackets after the first mention, and perhaps give a very small description as well.

Line 94 - Were the authors able to remove women who used opioids illicitly during the pregnancy or just those on a treatment register?

Line 127-128 - Should this be 360 to 91 days before conception (not 89)? Otherwise there is a one-day cross-over with the wash-out period.

Line 168 - Suggest "birthweight 2 standard deviations below the expected fetal weight for gestational age or lower", otherwise sounds like it is less than 2 standard deviations, e.g. 1 SD.

Line 193 - Doesn't the null difference between POA-exposed and acetaminophen-exposed offspring only provide support for confounding by indication if it is assumed that acetaminophen is safe and has no influence on the adverse outcomes? I think this needs to be discussed, perhaps in the Methods, with references to support this assumption.

Line 214 - 'exposures' should be 'exposure'.

Line 216 - what was the reason/justification for the restricted exposure window excluding 3 days before birth?

Line 235-236 - "Fifth, in order to assess if results with exposure in a single trimester and exposure in multiple trimesters would replicate with another exposure also evaluating quantity of exposure,…" I found this phrase difficult to follow. Perhaps "Fifth, in order to assess whether the single versus multiple trimester dose-response results would be replicated with another indicator based on quantity of exposure,…"

Line 259 - Does the inclusion of 9201 unique mothers mean that siblings with the same father but different mother were included? Can the authors elaborate on why these siblings were included? While siblings with the same father may account for some factors, they do not account for the mother-specific confounders, including ongoing health issues that might affect placental health and risk of preterm, SGA etc. 

Line 265 - Does the prescribed opioid analgesic exposure group exclude those prescribed acetaminophen?

Line 332 - Include short description of continuous outcome findings.

Line 349-351 - Aren't the commensurate findings the case for all analyses except for the sibling and cousin comparisons, which showed null findings?

Reviewer #3: This is an interesting study exploring the relationships between prescription opioid analgesia prescriptions and two birth outcomes, mainly preterm birth (defined as prior to 37 weeks) and small for gestational age (as a proxy for IUGR). The authors used a large robust database and attempted by using multiple analysis to control for as many confounders as they could (including the main issue of confounding by indication, by comparing with acetaminophen prescription and sibling comparisons). This is a vast improvement on other studies which didn't attempt to control for these factors. The authors do a good job of explaining the rationale for the study in the introduction and reviewing the literature to date. The abstract reflects the main findings, the writing is clear, though a bit wordy. The discussion clearly delineates the strengths, weaknesses, and areas for further study. It would be good to use such a large database to look at birth defects.

I would suggest a few changes.

Table 2 is very long and some of the variables are quite granular. It is unclear if their initial logistic regression analysis included all of these covariates and found them all to be non-significant (it would be interesting to look at poverty and smoking separately) and then settled on pre-pregnancy exposure and sibling exposure as being the ones that settled out from the logistic regression (and thus redid the analysis using just those variables). I'm not a statistician so I cannot speak to the robustness of the statistical analysis. 

I was interested to see that 18-29% of the fathers had prior criminal involvement (if I'm reading that correctly). I think that would influence the outcomes, especially if interpersonal violence is involved.

Reference 46 looks like it was glitched in the referencing software and needs to be redone.

There's a typo on page 25 of the supplementary materials.

Reviewer #4: 

Thank you for the opportunity to review this manuscript. I found it very informative and extremely well written

My only minor suggestions are: 

Line 63 The estimate of 30% of pregnant women filling opioid prescriptions seems very high, please give more details of this study.

Line 70/71, can you put in % for use of POA, illicit and medication assisted treatment 

Main analysis:

There is a great deal of information here. Some subheadings would help here given the many models tested, same comment for sensitivity analysis.

[LINK]

---

## [Decision Letter · Decision Letter 1]

18 Oct 2019

Dear Dr. Sujan,

Thank you very much for re-submitting your manuscript "Maternal prescribed opioid analgesic use during pregnancy and associations with adverse birth outcomes: A population-based study" (PMEDICINE-D-19-02701R1) for review by PLOS Medicine.

I have discussed the paper with my colleagues and the academic editor and it was also seen again by reviewers. I am pleased to say that provided the remaining editorial and production issues are dealt with we are planning to accept the paper for publication in the journal.

[LINK]

We look forward to receiving the revised manuscript by Oct 25 2019 11:59PM. 

Sincerely,

Louise Gaynor, MBBS PhD

Associate Editor 

PLOS Medicine

plosmedicine.org

Requests from Editors:

- the "Background" subsection of the abstract could do with trimming as it’s very long and really just need to set the scene on your study. Please also provide p values with 95% Cis (and in the main text and tables) . 

- Author summary needs reformatting as it should be bullet points. 

- "important clinical implications" too many times in the paragraph at line 467 – please reduce. 

- The references need reformatting (initials after surnames); refs 7 & 49 appear to be incomplete and it should be 6 names then et al, per Vancouver style.

- The STROBE checklist needs to be broken out of the supplementary file into a separate document. 

- Incidentally, the supplementary file has a typo ("conducing" -> "conducting", in line 4)

Comments from Reviewers:

Reviewer #1: Thanks authors for their effort to improve the manuscript. I am satisfied with the response and the revision. All my questions were well addressed. No further issues needing attention.

Reviewer #2: Thank you to the authors for responding to my concerns. I have no further questions.

[LINK]

---

## [Editor Report · Decision Letter 2]

4 Nov 2019

Dear Dr. Sujan, 

On behalf of my colleagues and the academic editor, Dr. Jenny Myers, I am delighted to inform you that your manuscript entitled "Maternal prescribed opioid analgesic use during pregnancy and associations with adverse birth outcomes: A population-based study" (PMEDICINE-D-19-02701R2) has been accepted for publication in PLOS Medicine. 

PRODUCTION PROCESS

PRESS

PROFILE INFORMATION

Thank you again for submitting the manuscript to PLOS Medicine. We look forward to publishing it. 

Best wishes, 

Clare Stone, PhD

Managing Editor 

PLOS Medicine

plosmedicine.org